


# SPATIALLY VARYING RELEVANCE OF HYDROMETEOROLOGICAL HAZARDS FOR VEGETATION PRODUCTIVITY EXTREMES

Josephin Kroll[1*], Jasper M. C. Denissen[1*], Mirco Migliavacca[1], Wantong Li[1], Anke Hildebrandt[2,3,4], Rene Orth[1]

* Authors contributed equally

– Department of Biogeochemical Integration, Max Planck Institute for Biogeochemistry, Jena, Germany
– German Centre for Integrative Biodiversity Research, Halle-Jena-Leipzig, Leipzig, Germany
- Helmholtz Centre for Environmental Research–UFZ, Leipzig, Germany
- Friedrich-Schiller-University Jena, Jena, Germany

Correspondence: Josephin Kroll (jkroll@bgc-jena.mpg.de) and Jasper M. C. Denissen (jdenis@bgc-jena.mpg.de)

## ABSTRACT

Vegetation plays a vital role in the Earth system by sequestering carbon, producing food and oxygen, and providing evaporative cooling. Vegetation productivity extremes have multi-faceted implications, for example on crop yields or the atmospheric CO2 concentration. Here, we focus on productivity extremes as possible impacts of coinciding, potentially extreme hydrometeorological anomalies. Using monthly global satellite-based Sun-induced chlorophyll fluorescence data as a proxy for vegetation productivity from 2007 - 2015, we show that vegetation productivity extremes are related to hydrometeorological hazards as characterized through ERA5-Land reanalysis data in approximately 50% of our global study area. For the latter, we are considering sufficiently vegetated and cloud-free regions; and we refer to hydrometeorological hazards as water or energy related extremes inducing productivity extremes. The relevance of the different hazard types varies in space; temperature-related hazards dominate at higher latitudes with cold spells contributing to productivity minima and heat waves supporting productivity maxima, while water-related hazards are relevant in the (sub)tropics with droughts being associated with productivity minima and wet spells with the maxima. Next to single hazards also compound events such as joint droughts and heat waves or joint wet and cold spells play a role, particularly in dry and hot regions. Further, we detect regions where energy control transitions to water control between maxima and minima of vegetation productivity. Therefore, these areas represent hot spots of land-atmosphere coupling where vegetation efficiently translates soil moisture dynamics into surface fluxes such that the land affects near-surface weather. Overall, our results contribute to pinpoint how potential future changes in temperature and precipitation could propagate to shifting vegetation productivity extremes and related ecosystem services.

## 1 INTRODUCTION

Vegetation is a crucial component of the Earth system because it provides ecosystem services like food and oxygen production, $CO_2$ sequestration and evaporative cooling. Therefore, the effects of changes in vegetation productivity are diverse; it influences crop yields (Orth et al., 2020), cloud formation (Hong



et al., 1995; Freedman et al., 2001), precipitation (Pielke Sr et al., 2007), atmospheric pollution (Otu-
Larbi et al., 2019) and heat wave intensity (Li et al., 2021b).
Photosynthesis requires sufficient water (soil moisture) and energy (incoming shortwave radiation)
supply. In regions that are water (energy) limited, plants usually benefit from water (energy) surpluses
and suffer from respective deficits. Many studies confirm that, depending on the evaporative regime,
vegetation productivity follows the temporal evolution of influential variables such as soil moisture or
temperature which summarize the water or energy dynamics (Beer et al., 2010; Seddon et al., 2016;
Madani et al., 2017; Denissen et al., 2020; Piao et al., 2020; Li et al., 2021a).
Correspondingly, hydrometeorological hazards, such as temperature and precipitation extremes have
implications on vegetation productivity. Many studies investigated the influence of such hazards on
vegetation productivity, highlighting their impact on the biosphere (Ciais et al., 2005; Zhao et al., 2010;
Zscheischler et al., 2013; Zscheischler et al., 2014a; Zscheischler et al., 2014b; Flach et al., 2018; Wang
et al., 2019; Zhang et al., 2019; Qui et al., 2020). However, usually these studies focus on particular types
of hydrometeorological hazards such as droughts or heat waves, or they use vegetation productivity
data from models or other proxies rather than the recent satellite-derived Sun-induced chlorophyll
fluorescence (SIF) data (Frankenberg et al., 2011; Joiner et al., 2013).
In this study, we re-visit the relationship between vegetation productivity and hydrometeorological
hazards by, to our knowledge, for the first time comprehensively analyzing the implications of both
single and compound hazards on observation-based vegetation productivity extremes as inferred from
SIF data across the globe. This analysis is done from an impact perspective; we first detect impacts
(productivity extremes) before relating them to coinciding, potentially extreme hydrometeorological
anomalies (Smith, 2011). Finally, we investigate where the full vegetation productivity range between
minima and maxima involves transitions from energy to water controls. In regions where this occurs,
the feedback of the land surface on the climate can be stronger, as the water-controlled vegetation
translates soil moisture dynamics through its energy and water fluxes to affect the boundary layer and
consequently also near-surface weather. Hence, our vegetation-based analysis can indicate hot spots
of land-atmosphere coupling (Koster et al., 2004; Guo and Dirmeyer, 2013).
In section 3.1 we investigate the co-occurrence of vegetation productivity extremes and
hydrometeorological hazards. Further, we show the timing of such vegetation productivity extremes in
section 3.2. Additionally, we determine the main drivers of vegetation productivity extremes and assess
the influence of underlying evaporative regimes in section 3.3. We summarize our results across climate
regimes in section 3.4 and investigate regions with vegetation productivity controls switching between
water and energy variables in section 3.5.

## 2  DATA AND METHODS

In order to characterize vegetation behavior, we use SIF and Normalized Difference Vegetation Index
(NDVI) data in this study. SIF is used as a proxy for vegetation productivity. We employ satellite-observed
SIF data retrieved from the Global Ozone Measurement Experiment (GOME-2; Koehler et al., 2015). In
the derivation of this SIF product, multiple corrections for varying solar zenith angles, differences in
overpass times and cloud fraction have been applied to yield reliable SIF estimates. In addition to
vegetation productivity, we also study changes related to vegetation greenness by using satellite-
observed Normalized Difference Vegetation Index (NDVI) data from Moderate-resolution Imaging
Spectroradiometer (MODIS; Didan, 2015).
As for the hydrometeorological variables, representing energy and water availability, we consider 2m
temperature, shortwave incoming radiation, vapor pressure deficit, soil moisture from 4 layers (1: 0-7
cm, 2: 7-28 cm, 3: 28-100 cm, 4: 100-289 cm) and total precipitation, all from the ERA5-Land reanalysis
data (Muñoz-Sabater, 2019). In addition to this, and to validate the robustness of our results, we use an
alternative soil moisture product, SoMo.ml, which provides data for three layers (1: 0-10 cm, 2: 10-
30cm, 3: 30-50cm), and which is derived through a machine learning approach that is trained with in-
situ soil moisture measurements from across the globe (O and Orth, 2021). All datasets used in this
study are summarized in Table 1.





Table 1. Data sets used in this study.

| Variables | Dataset | Version | Application | Reference |
|---|---|---|---|---|
| Sun-induced chlorophyll fluorescence | GOME-2 | GFZ | Vegetation productivity proxy | Köhler et al., 2015 |
| Normalized difference vegetation Index | MOD13C1 | V006 | Vegetation greenness proxy | Didan, 2015 |
| Soil moisture layer 1-4, precipitation, shortwave incoming radiation, temperature, vapor pressure deficit | ERA5 land | | Hydrometeorological variables indicating energy and water availability | Muñoz-Sabater, 2019 |
| Precipitation, net solar radiation, net thermal radiation | ERA5 | | Computation of aridity to evaluate resulting patterns | Hersbach et al., 2020 |
| Soil moisture layer 1-3 | SoMo.ml | 1 | Alternative soil moisture data set | O and Orth, 2021 |
| Fraction of vegetation cover | VCF5KYR | 1 | Evaluation of resulting patterns with respect to vegetation characteristics | Hansen and Song, 2018 |

The workflow applied to these datasets is illustrated in Fig. 1. At first, all data is pre-processed for
comparability by (i) aggregating it to monthly, half-degree spatial and temporal resolution and by (ii)
focusing on the time period 2007-2015. Next, we compute anomalies by removing linear trends and the
mean seasonal cycle from the data for both the vegetation and hydrometeorological variables. In each
grid cell, we disregard months with an absolute SIF value below 0.5 mW/m$^2$/sr/nm to focus on times
with sufficiently active vegetation (as in Li et al., 2021a). Additionally, grid cells with a fractional
vegetation cover < 5% are excluded from the analysis. Finally, we assure the necessary data availability
by considering only grid cells with > 15 monthly anomalies across the study period remaining after the
filtering. Out of the identified suitable months in each grid cell, we determine the five strongest negative
and five strongest positive monthly SIF anomalies. The sum of all grid cells for which five SIF maxima
and minima can be detected is referred to as total study area.
After this filtering, we follow two approaches in our analysis. In the first approach, we check for
hydrometeorological hazards coinciding with the determined extreme vegetation productivity events.
Thereby, we consider air temperature and soil moisture layer 2 as these variables were previously found
to be globally most relevant for vegetation productivity (Li et al., 2021a). At first we average the monthly
temperature and soil moisture anomalies across the five months of maximum and minimum SIF
anomalies. Then, a series of steps is taken to test if the coinciding hydrometeorological anomalies during
SIF extremes are actually hazardous: (i) We randomly sample five months with sufficiently active
vegetation and average the soil moisture and temperature anomalies, respectively, across them. (ii) We
repeat this 100 times to obtain a distribution from which we determine the 10$^{th}$ and 90$^{th}$ percentile. (iii)
A hydrometeorological hazard is detected if the actual, averaged temperature and/or soil moisture
anomalies associated with the SIF extremes are below 10$^{th}$ (cold spell or drought) or above the 90$^{th}$
percentile (heat wave or wet spell) of the distribution of randomly sampled averaged anomalies. Note
that with this approach we can detect both single and compound hydrometeorological hazards.
Complementing this analysis, in the second approach we analyze the temporal co-variation between SIF
extremes and hydrometeorological anomalies. For this purpose, we correlate the five SIF extreme
anomalies with anomalies of all considered hydrometeorological variables in each grid cell. We include





respective SIF and hydrometeorological data from the surrounding grid cells to yield a larger data
sample consisting of 5 x (8+1) = 45 data pairs. We disregard negative and insignificant (p-value > 0.05)
correlations, as we assume these are not indicating actual physical controls but rather represent the
influence of noise or confounding effects such as low precipitation during times of high radiation. Finally,
the hydrometeorological variable that yields the highest correlation coefficient with the extreme SIF
anomalies is regarded as the main SIF-controlling variable during vegetation productivity maxima or
minima.

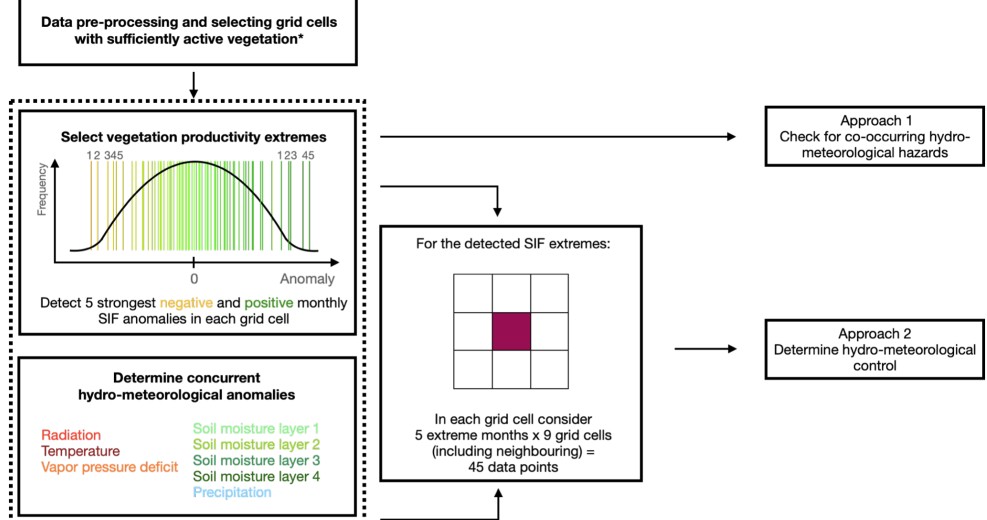

Figure 1. Schematic representation of our methodological approach. *Filtering for sufficiently active vegetation is explained
in section 2.

# 3  RESULTS AND DISCUSSION

## 3.1  HYDROMETEOROLOGICAL HAZARDS AND VEGETATION PRODUCTIVITY EXTREMES

Figure 2 shows which hydrometeorological hazards are associated with SIF extremes as inferred with
approach 1 described in Section 2 and in Fig. 1. In approximately 50% of the global study area, we find
that vegetation productivity extremes are associated with hydrometeorological hazards. This is in line
with previous research (Zscheischler et al., 2014b). For both maximum and minimum vegetation
productivity, we find spatially coherent patterns of associated hydrometeorological hazards. In the
Northern Hemisphere SIF maxima (minima) at high latitudes relate to heat waves (cold spells), where in
mid latitudes they occur jointly with wet spells (droughts). This suggests that hydrometeorological
hazards associated with SIF extremes vary systematically according to energy- and water control of the
local vegetation. Thereby, the boundary between both regimes and the respectively determined
relevant hydrometeorological hazards is surprisingly sharp, for example in North America, and in eastern
Europe and Russia (Flach et al., 2018).
Further, single hydrometeorological hazards (either an extreme temperature or soil moisture anomaly)
are relevant in more areas than compound hazards (combination of extreme temperature and extreme
soil moisture anomaly). Compound hazards seem to be particularly important in the sub-tropics on both
hemispheres. Differences also exist between maximum and minimum vegetation productivity extremes,
the latter being slightly more associated with compound hazards.
Overall, the most frequent hazards during vegetation productivity minima are droughts and cold spells.
Previous studies have reported the relevance of drought in this context (Zscheischler et al., 2013;
Zscheischler et al., 2014a; Zscheischler et al., 2014b) even though for different vegetation productivity




proxies. On the contrary, the importance of cold spells is not analyzed, probably because vegetation
productivity in boreal regions is comparably smaller than in e. g. tropical regions (Li and Xiao, 2019).
The results in Fig. 2 are based on averages of the five months with strongest SIF anomalies in each grid
cell. Figure S1 shows co-occurring hydrometeorological hazards separately for each of the five SIF
maxima and minima. The patterns are similar as in Fig. 2, we consistently find temperature-related
hazards to be relevant in energy-controlled regions and water-related hazards in water-controlled
regions across all five individual SIF extremes. Weaker SIF extremes tend to be less associated with
hydrometeorological hazards. This could be because the signal-to-noise ratio is decreased for weaker
extremes, or other factors such as disturbances (fire or insect outbreaks) play a more prominent role
for these productivity extremes. As mentioned, soil moisture layer 2 is used here to detect droughts and
wet spells, but similar results are obtained with soil moisture layers 1 and 3, respectively (not shown).

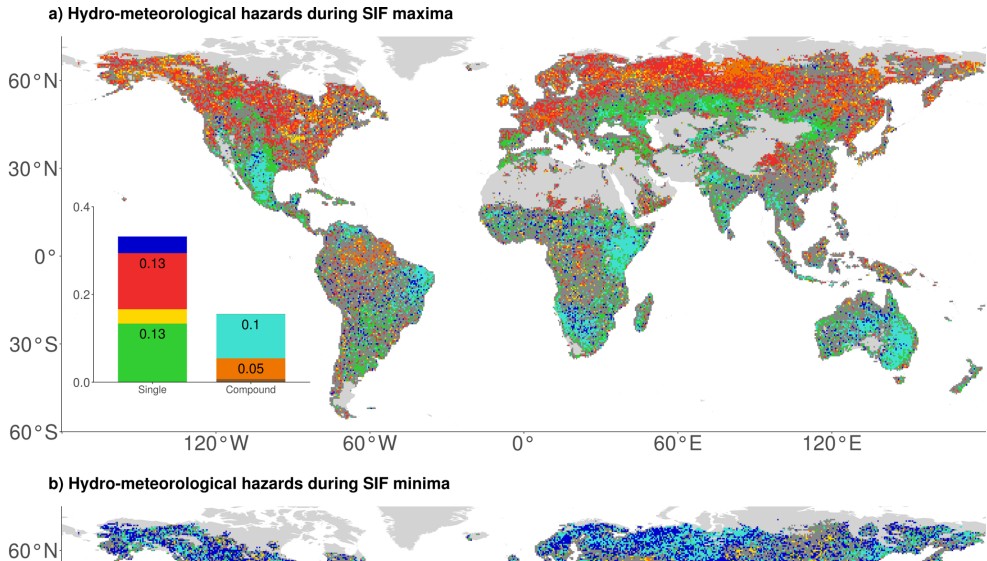

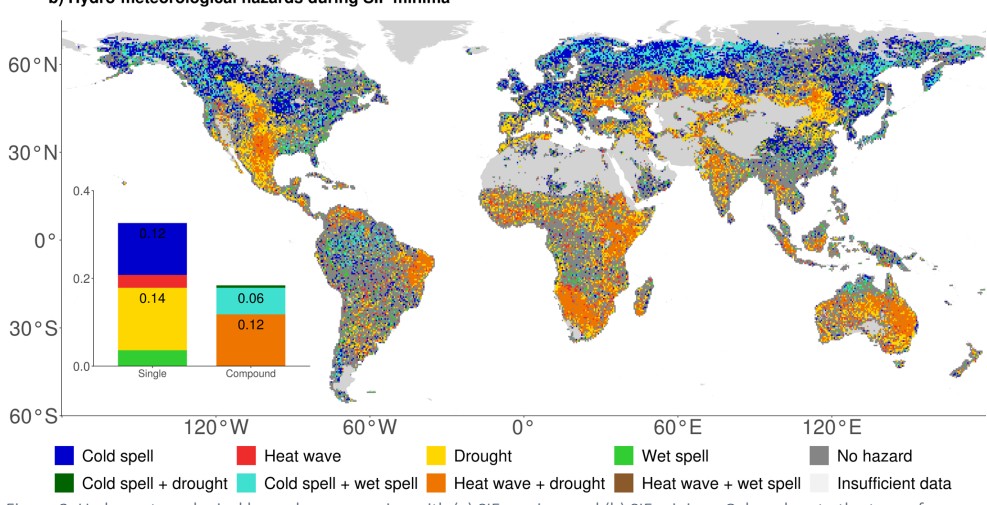

Figure 2. Hydrometeorological hazards co-occurring with (a) SIF maxima and (b) SIF minima. Colors denote the type of
hydrometeorological hazard. Bar plots indicate the area affected by each hazard type relative to the total study area.

3.2  TIMING OF STRONGEST SIF EXTREME



To further understand the spatially varying relevance of hydrometeorological hazards, we show the
months of the year associated with the strongest SIF extreme in each grid cell in Fig. 3. The spatial
pattern is quite different from that in Fig. 2, for example the sharp transitions between regions with
energy and water-related hydrometeorological hazards are not present in Fig. 3. Hence, this transition
is apparently not related to SIF extremes occurring in different seasons and might be rather related to
different evaporative regimes which will be further investigated in the next subsection 3.3. The spatial
variability in Fig. 3 is lower at high latitudes compared with (sub-)tropical regions. At high latitudes the
growing season is short and constrained by energy availability. In the tropics, we find an increased
smaller-scale variability, presumably due to the weak seasonal cycle of hydrometeorological variables.
Most SIF extremes in North America and Eurasia occur in the early growing season, presumably when
vegetation either starts to grow or growing is limited due to energy or water control. While here we
show the months-of-year associated with the strongest SIF extreme, in Fig. S2 we show similar patterns
in the timing of the 2$^{nd}$ to 5$^{th}$ strongest SIF extremes, indicating that each of the remaining SIF extremes
occurs in similar months-of-year.

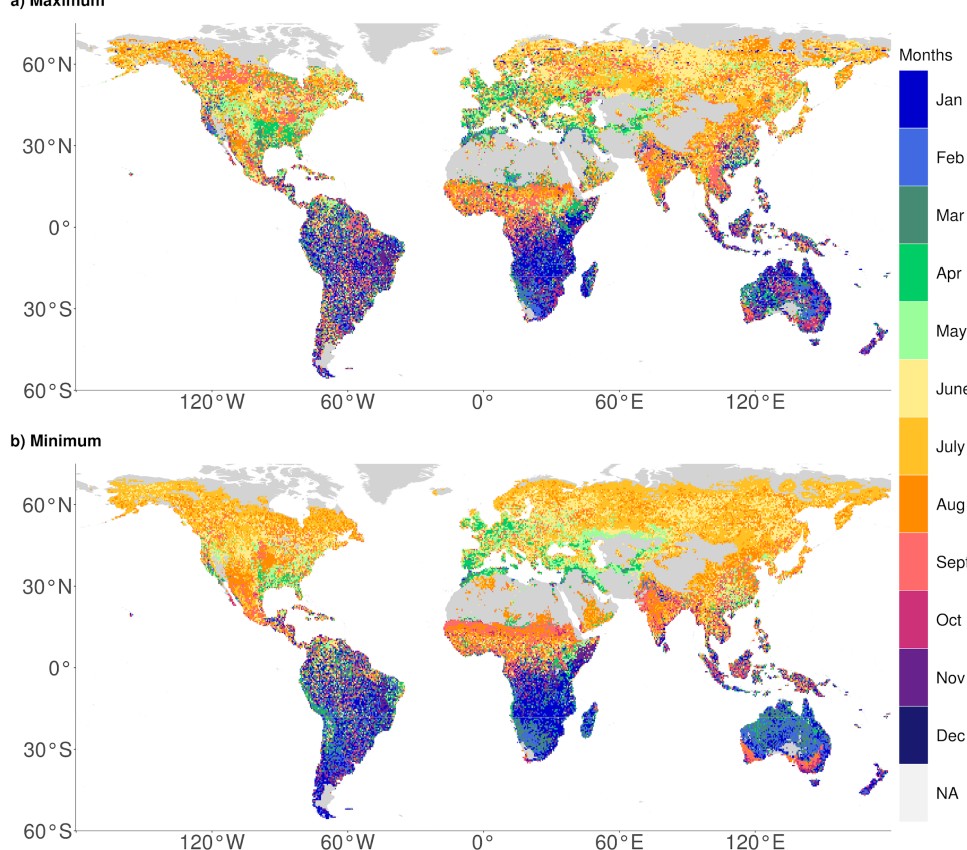

Figure 3. Global distribution of the month-of-year in which the strongest SIF (a) maximum and (b) minimum anomaly occur.
Data gaps (grey) are caused by filtering for active vegetation and excluding insignificant and negative correlations.
3.3  HYDROMETEOROLOGICAL DRIVERS OF VEGETATION PRODUCTIVITY EXTREMES
After showing the co-occurrence of hydrometeorological hazards with SIF extremes, we apply a
correlation analysis (approach 2 in section 2) to characterize the co-variability between extreme SIF
anomalies and concurrent hydrometeorological anomalies. Figure 4 shows the hydrometeorological



variable that correlates strongest with SIF during extreme vegetation productivity months, indicating
respective controls. At the high latitudes and in the tropics SIF extremes are generally energy controlled,
while in the mid latitudes and subtropics they are water controlled. Overall, we find similar spatial
patterns as in Fig. 2, demonstrating consistent results across co-occurrence and co-variability of SIF
extremes and hydrometeorological hazards. This coherence suggests that hydrometeorological hazards
play a key role in inducing SIF extremes.
The bar plot insets in Fig. 3 indicate that SIF maxima are predominantly controlled by energy variables
while SIF minima are overall more controlled by water variables. Even though weaker, this shift is also
present in Fig. 2. This difference can be explained with transitional regions, which have energy-
controlled SIF maxima, but water-controlled SIF minima. This is illustrated for example by the northward
shift of the transition between energy and water control in Russia when comparing the results for
maximum and minimum SIF. These transitional regions will be further investigated in section 3.5.
We repeated this analysis with SoMo.ml soil moisture and found similar spatial patterns of energy- and
water-controlled regions (Fig. S3), underlining that our results are robust with respect to the choice of
the soil moisture product. Furthermore, we repeat our co-variability analysis for NDVI instead of SIF in
Fig. S4, which allows us to contrast to some extent the behavior of vegetation physiology (SIF) and
vegetation structure (NDVI). Similar to the spatial patterns of energy- and water-controlled vegetation
in Fig. 4, NDVI shows predominant energy control at high latitudes, while the mid latitudes are largely
water-controlled. Further, as in Fig. 4 for SIF, NDVI minima are more associated with water variables
than NDVI maxima.
However, the overall extent of water-controlled areas is clearly larger in the case of NDVI compared
with the SIF results. This could (i) partly be related to the fact that NDVI, being less dynamic than SIF
because it is more related to vegetation greenness and structure, tends to vary at time scales more in
line with that of soil moisture (Turner et al., 2020), which can support stronger correlations. (ii) It could
be due to confounding effects of the changing soil/vegetation color between dry and wet states on the
NDVI signal. (iii) NDVI tends to saturate for canopies with high Leaf Area Index and tend to be relatively
stable over evergreen boreal forests (Turner et al., 1999; Walther et al., 2015). This can mask significant
correlations between NDVI and energy related variables in the boreal regions that are more energy
controlled (Fig. 4).



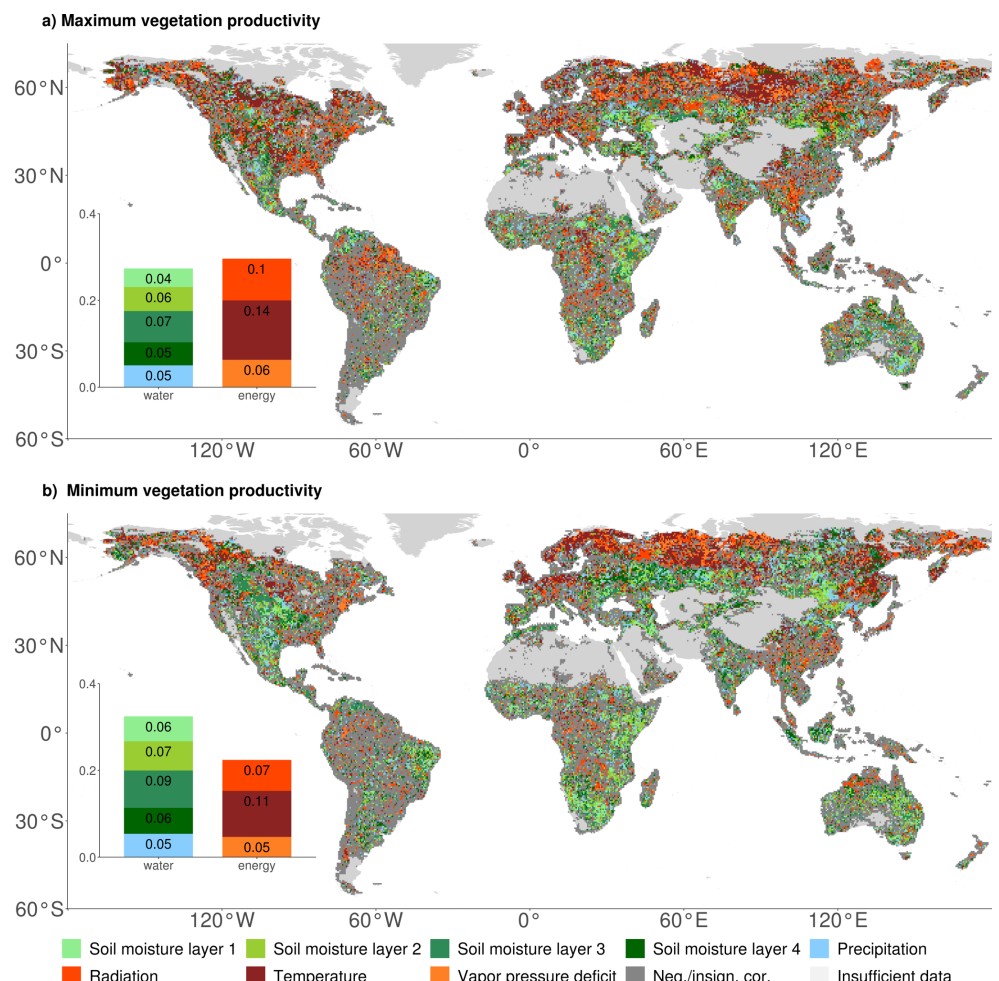

Figure 4. Global distribution of hydrometeorological controls of Sun-Induced Fluorescence (SIF) (a) maxima and (b) minima in
respective colors, as assessed from strongest correlations. The inset bar plot indicates the area controlled by each variable
relative to the total study area. Dark grey color denotes the study area, in which correlations are negative/insignificant.

### 3.4 HYDROMETEOROLOGICAL CONTROLS ACROSS CLIMATE REGIMES

In addition to analyzing the spatial variation of the main drivers of vegetation productivity extremes, we
attempt to further understand the large-scale patterns along temperature and aridity gradients. To this
end, we bin grid cells by their climate characteristics as denoted by long-term mean temperature and
aridity (the ratio between unit-adjusted net radiation and precipitation). The results in Fig. 5 illustrate
which hydrometeorological variable most often has the highest correlation with SIF anomalies in each
climate regime.
Figure 5 (a) and (b) show that vegetation productivity extremes in humid regions (aridity < 1; Budyko,
1974) are mostly energy controlled, with temperature controlling in cold regions (long-term average
temperature < 10 °C) and radiation controlling in warm regions (long-term temperature > 10 °C). In
contrast, productivity extremes in arid regions (aridity > 2, Budyko, 1974) are mainly water controlled,
with soil moisture layer 2 and 3 as most important water controls. The main difference between
maximum and minimum SIF results is detectable in semi-arid regions (1 < aridity < 2). While for
maximum SIF those climate regimes show mostly energy control, SIF minima in these regimes are largely
water controlled. From this, we deduce that semi-arid regions represent the transitional regime, as the





main drivers change from energy to water variables from SIF maximum to SIF minimum.
The results for NDVI show similar patterns despite an increased overall water control as seen earlier in
the global maps (Fig. S4). For example, where in humid regions SIF extremes are mainly energy
controlled, NDVI extremes are more often water controlled, which is also reflected in the global maps
in Fig. S4.
Figure 5 (e) and (f) show the results of Fig. 2 binned according to their long-term climate characteristics.
In humid regions, both SIF extremes are co-occurring with temperature hazards. In contrast, in arid
regions water-related hazards co-occur with maximum and minimum SIF. Thereby, Fig. 5 underlines
once more the similarity of the results obtained with approaches 1 (Fig. 2) and 2 (Fig. 4).
To additionally explore the influence of different vegetation types on the main controls of vegetation
productivity, we bin the grid cell results by the respective fraction of tree cover of the entire vegetation
cover, and by aridity in Fig. S5. We find that the radiation control of SIF extremes in humid regions is
mostly associated with forests, and that the water control in semi-arid regions largely occurs for shorter
vegetation while productivity extremes in more forested semi-arid regions tend to be energy-controlled.
As in Fig. 5, similar patterns are found for NDVI extremes with overall increased relevance of water
variables particularly in short vegetation-dominated regions.







Figure 5. Hydrometeorological controls of vegetation productivity extremes summarized across climate regimes, (a) and (b)
for Sun-Induced Fluorescence (SIF) extremes, (c) and (d) for Normalized Difference Vegetation Index (NDVI) extremes. (e) and
(f) display the hydrometeorological hazards co-occurring with the SIF extremes. Box color denotes the main controlling



hydrometeorological variable, the second most important variable is indicated in the smaller squares' color, while its size
represents the ratio between highest/second highest amounts of grid cells.
3.5    SWITCHING HYDROMETEOROLOGICAL CONTROLS BETWEEN SIF MAXIMA AND MINIMA
In a final step, we focus on shifts between energy and water control when moving from SIF maxima to
SIF minima. The respective transitional regions represent hot spots of land-atmosphere coupling as (i)
in these regions the land surface (soil moisture) is affecting near-surface weather at least during
productivity minima (therefore also influencing transpiration) and (ii) this effect can be significant as
transpiration (variability) is relatively high compared with drier regions where vegetation productivity
would be water-limited across its entire range from minimum to maximum. The results are depicted in
Fig. 6, which illustrates these emerging transitions from water to energy control (yellow) and vice-versa
(blue, denoting land-atmosphere hot spots). Grid cells that stay within water or energy control, even
with a change between the water or energy variables, respectively, are shown in black indicating no
transition. Figure 6 (a) reveals many regions with no transition. Transitions are found mostly in North
Eurasia and North America. Globally, a change from energy control during maximum SIF to water control
during minimum SIF occurs more often (8% of the study area) than the opposite transition (5%).
Figure 6b and c display the percentage of grid cells in each climate regime changing from water to
energy control and vice-versa with grid cells binned with respect to long-term climate conditions, similar
to Fig. 5. The highest fraction of grid cells in each climate regime would show no change, but as we focus
on transitioning grid cells, only they are displayed. Transitions from water to energy control between
SIF maxima and SIF minima happen most often in cold, humid regions. This is deviating from the
prevailing energy control in these climate regimes, and probably related to local-scale features and/or
micro-meteorological conditions. Figure 6 (c) indicates that changes from energy control during
maximum SIF to water control during minimum SIF most frequently occur in the semi-arid transitional
regions. These are land-atmosphere coupling hot spots as described above. The transition from energy
to water limitation could be caused by energy-controlled maxima in spring, when presumably soil water
resources are available after being replenished during autumn and winter. With sufficient water supply,
energy surpluses could induce vegetation productivity maxima. During summer, soil moisture could be
depleted for example by the high vegetation demand, and therefore taking over the SIF control of
photosynthesis that is reflected into the SIF dynamics.



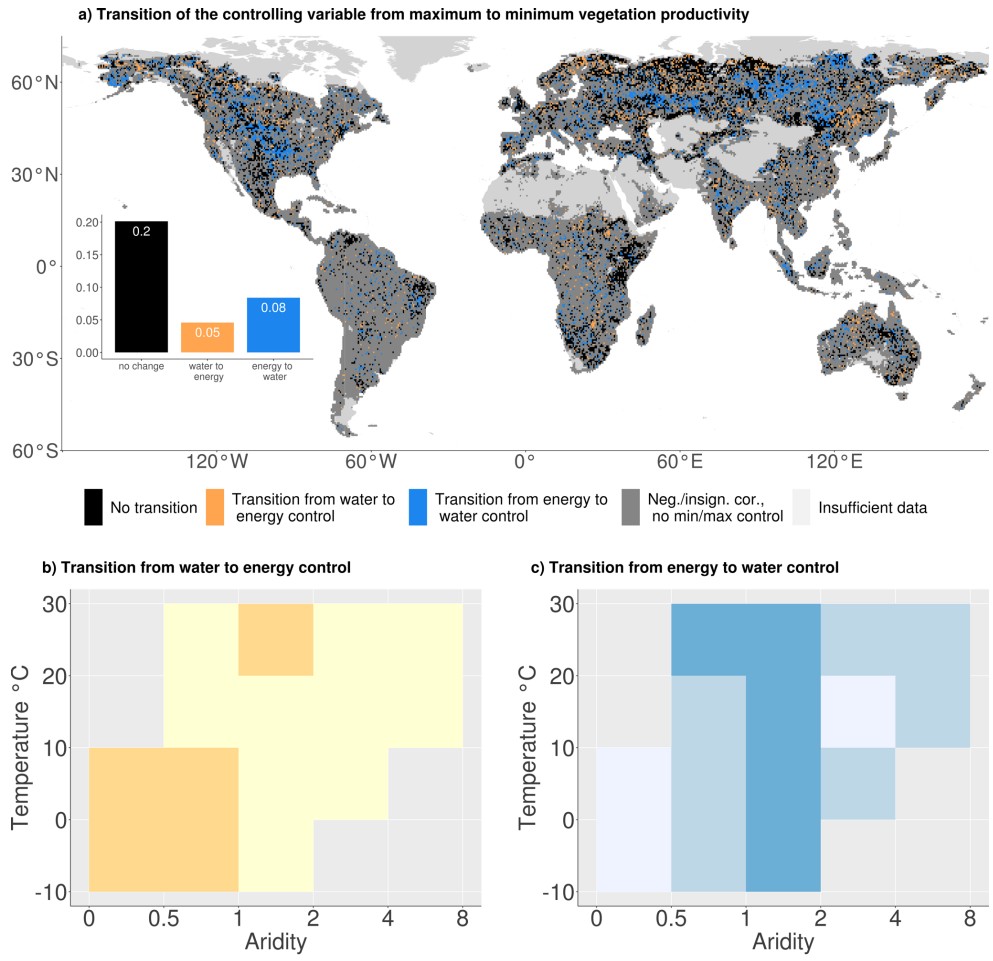

Figure 6. Changing hydrometeorological controls between vegetation productivity maxima and minima. (a) Global distribution of changing controls: In Fig. (b) and (c) grid cells are binned by their long-term climate characteristics. (b) indicates the percentage of grid cells in each climate regime switching from water to energy control, (c) denotes the percentage of grid cells changing from an energy-controlled maxima to a water-controlled minima.

## 3.6 LIMITATIONS

Our results are obtained at, and valid for, relatively large spatial (half degree) and temporal (monthly) scales. Previous studies have shown differences in the vegetation-climate coupling across scales (Linscheid et al., 2020), suggesting it would be worthwhile to repeat our analysis for different spatiotemporal scales in the future, possibly with new satellite data products. In this context it should be noted, however, that while the relationship between SIF and gross primary productivity (GPP) as actual vegetation productivity is strong for large spatio-temporal scales (Frankenberg et al., 2011; Joiner et al., 2013), it can deteriorate towards smaller scales (He et al., 2020; Maguire et al., 2020; Marrs et al., 2020). And the spatiotemporal range within which there is an acceptable SIF-GPP relationship is not entirely clear yet.

As a second source of uncertainty, SIF data with their relatively large spatial footprint are more vulnerable to cloud contamination compared to finer-scale satellite products (Joiner et al., 2013). Also, especially across South America the SIF data quality is decreased to additional noise (Joiner et al., 2013; Köhler et al., 2015). In our study, many grid cells in these regions and other tropical, cloud-dominated



regions exhibit insignificant or negative correlations between SIF and hydrometeorological anomalies,
which is why no hydrometeorological controls can be determined there (Fig. 4). Confirming the validity
of our results for the tropical grid cells where results can be obtained, we find mostly consistent and
physically meaningful results, e. g. radiation being a main driver of vegetation productivity as the cloud
cover is limiting radiation (reported similarly for non-extreme conditions by Green et al., 2020 and Li et
al., 2021a).
Next to the SIF data, there is also noteworthy uncertainty in the soil moisture data from ERA5. While
data quality of surface soil moisture benefits from (satellite) data assimilation, the soil moisture
dynamics in deeper layers are more model-based which is somewhat contradicting the observational
character of our study. Therefore, we use soil moisture data from SoMo.ml as an independent data set,
which is not based on physical modelling and the related assumptions and parameterizations as it is
derived with machine learning applied to in situ measurements from different depths. Overall, the
similar results obtained with ERA5-Land and SoMo.ml soil moisture confirm the robustness of our results
despite uncertainties in the soil moisture data.
Finally, the use of correlation methods for inferring causal relations is potentially insufficient and under
debate (Krich et al., 2020). We want to emphasize that in our study when referring to "drivers" or
"controls" of vegetation productivity, we simply base this on correlation and do not imply causality.
Nevertheless, we try to filter out confounding effects by disregarding negative and insignificant
correlations. Additionally, testing our methodology (approach 2) for non-anomalous vegetation
productivity (Fig. S6) which allows to compare results with that of Li et al. (2021a), reveals similar results
while they use a different methodology based on random forests and Shapley Additive Explanations
(SHAP) values which is more robust against confounding effects. Next to this, in our study we apply two
different methodologies in approaches 1 and 2 and find similar results, which further underlines the
robustness of our conclusions.

## 4  CONCLUSION

In this observation-based study, we quantify that vegetation productivity extremes are related to
hydrometeorological hazards in about 50% of the global land area that is sufficiently vegetated and
cloud-free. The most relevant hazards for vegetation productivity extremes vary along climate
gradients. For vegetation productivity maxima the most relevant hydrometeorological extremes are
heatwaves in Northern latitudes above 50°N and wet spells in latitudes below 50°N. For productivity
minima, drought and cold spells are globally most detrimental to large-scale photosynthesis and carbon
uptake. The results of our impact-centric analysis are similar to, and complement more traditional
climate-centric studies (Ciais et al., 2005; Flach et al., 2018; Qui et al., 2020). Compound extremes also
play a role in 15-20% of our study area, they are somewhat more relevant for productivity minima than
for the maxima, with joint drought-heat extremes being most important. Semi-arid, grass-dominated
ecosystems tend to transition between water and energy control within the range of their productivity
variability. This results in a sensitivity to both water- and energy-related hazards. Thereby, we illustrate
how global land-atmosphere coupling hot spots (Koster et al., 2004), where the land surface affects
near-surface weather, can be verified using novel vegetation productivity data.
Overall, this study highlights the profound role of (compound) hydrometeorological hazards for global
vegetation productivity extremes. Understanding these complex, climate-dependent relationships with
present-day observational data is a starting point to more reliably foresee respective changes in a
changing future climate with e. g. less cold spells but probably more droughts.

## ACKNOWLEDGEMENTS

The authors thank Ulrich Weber for help with obtaining and processing the data. W. Li acknowledges
funding from a PhD scholarship from the China Scholarship Council. J.M.C. Denissen, J. Kroll, and R. Orth
acknowledge funding by the German Research Foundation (Emmy Noether grant number 391059971).





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
