# Peer review of "Spatially varying relevance of hydrometeorological hazards for vegetation productivity extremes"

_Biogeosciences, 2021_

## Author Comment (AC2)

**Supplement on RC**

[Figure]

Figure 1*: Hydrometeorological controls (ERA5 land) of different climate regimes on VHI from NOAA (Kogan et al., 1995). Grid cells are grouped by their long-term temperature and aridity (unit-adjusted net radiation/precipitation). The variable which is important for most of the grid cells for vegetation productivity maxima (a) and minima b), inferred using ET, in one climate regime is used to color the box. The second most important variable colors the smaller squares. Their ratio is denoted in the size of the squares.

**Main manuscript figures**

[Figure]

**a) Maximum vegetation productivity**

[Figure]

**b) Minimum vegetation productivity**

[Figure]

Figure 4. Global distribution of hydrometeorological controls of Sun-Induced Fluorescence (SIF) (a) maxima and (b) minima in respective colors, as assessed from strongest correlations. The inset bar plot indicates the area controlled by each variable relative to the total study area. Dark grey color denotes the study area, in which correlations are negative/insignificant.

[Figure]

Legend:
- Soil moisture layer 1
- Soil moisture layer 2
- Soil moisture layer 3
- Soil moisture layer 4
- Precipitation
- Radiation
- Temperature
- Vapor pressure deficit
- Neg./insign. cor.
- Insufficient data

a) SIF, Maximum

b) SIF, Minimum

c) EVI, Maximum

d) EVI, Minimum

e) Hazards, Maximum

f) Hazards, Minimum

Legend:
- Cold spell
- Heat wave
- Drought
- Wet spell
- No hazard
- Cold spell + drought
- Cold spell + wet spell
- Heat wave + drought
- Heat wave + wet spell
- Insufficient data

Importance 2/Importance 1  · < 80%  ■ 80% - 90%  ■ > 90%

Figure 5. Hydrometeorological controls of vegetation productivity extremes summarized across climate regimes, (a) and (b) for Sun-Induced Fluorescence (SIF) extremes, (c) and (d) for Enhanced Vegetation Index (EVI) extremes. (e) and (f) display the hydrometeorological hazards co-occurring with the SIF extremes. Box color denotes the main controlling hydrometeorological variable, the second most important variable is indicated in the smaller squares' color, while its size represents the ratio between highest/second highest amounts of grid cells.

**Supplementary material figures**

[Figure]

Fig. S4. Hydrometeorological controls (ERA5 land) of different climate regimes with a lag time of 1 month. Grid cells are grouped by their long-term temperature and aridity (unit-adjusted net radiation/precipitation). The hydrometeorological variables of the month preceding the SIF extreme have been used in the computation of most important variable. The variable which is important for most of the grid cells for vegetation productivity maxima (a) and minima b), inferred using SIF, in one climate regime is used to color the box. The second most important variable colors the smaller squares. Their ratio is denoted in the size of the squares.

[Figure]

a) Maximum vegetation productivity

b) Minimum vegetation productivity

Soil moisture layer 1   Soil moisture layer 2   Soil moisture layer 3   Soil moisture layer 4   Precipitation

Radiation   Temperature   Vapor pressure deficit   Neg./insign. cor.   Insufficient data

Fig. S5. Global distribution of hydrometeorological controls (ERA5 land) of EVI (a) maxima and (b) minima. The displayed variable correlates strongest with EVI in the extreme months, considering only significant and positive correlations. The bar plot indicates the area controlled by each variable relative to the total study area.

[Figure]

Fig. S6. Hydrometeorological controls (ERA5 land) of different climate regimes on ET from GLEAM. Grid cells are grouped by their long-term temperature and aridity (unit-adjusted net radiation/precipitation). The variable which is important for most of the grid cells for vegetation productivity maxima (a) and minima b), inferred using ET, in one climate regime is used to color the box. The second most important variable colors the smaller squares. Their ratio is denoted in the size of the squares.

[Figure]

Fig. S7. Hydrometeorological controls (ERA5 land) of different vegetation regimes. Grid cells are grouped by their fraction of tree cover and aridity (unit-adjusted net radiation/precipitation). The variable which is important for most of the grid cells for vegetation productivity extremes (a) and b) SIF; c) and d) EVI) in one vegetation regime is used to color the box. The second most important variable colors the smaller squares. Their ratio is denoted in the size of the squares.

**References**

Joiner, J., Guanter, L., Lindstrot, R., Voigt, M., Vasilkov, A. P., Middleton, E. M., ... & Frankenberg, C. (2013). Global monitoring of terrestrial chlorophyll fluorescence from moderate-spectral-resolution near-infrared satellite measurements: methodology, simulations, and application to GOME-2. Atmospheric Measurement Techniques, 6(10), 2803-2823. https://doi.org/10.5194/amt-6-2803-2013

Jonard, F., De Cannière, S., Brüggemann, N., Gentine, P., Short Gianotti, D. J., Lobet, G., Miralles, D. G., Montzka, C., Pagán, B. R., Rascher, U., & Vereecken, H. (2020). Value of sun-induced chlorophyll fluorescence for quantifying hydrological states and fluxes: Current status and challenges. Agricultural and Forest Meteorology, 291(June), 108088. https://doi.org/10.1016/j.agrformet.2020.108088

Karnieli, A., Bayasgalan, M., Bayarjargal, Y., Agam, N., Khudulmur, S., & Tucker, C. J. (2006). Comments on the use of the vegetation health index over Mongolia. International Journal of Remote Sensing, 27(10), 2017-2024. https://doi.org/10.1080/01431160500121727

Kogan, F. N. (1995). Application of vegetation index and brightness temperature for drought detection. Adv. Sp. Res. 15, 91–100. https://doi.org/10.1016/0273-1177(95)00079-T

Köhler, P., Frankenberg, C., Magney, T. S., Guanter, L., Joiner, J., & Landgraf, J. (2018). Global retrievals of solar-induced chlorophyll fluorescence with TROPOMI: First results and intersensor comparison to OCO-2. Geophysical Research Letters, 45(19), 10-456. https://doi.org/10.1029/2018GL079031

Sun, Y., Fu, R., Dickinson, R., Joiner, J., Frankenberg, C., Gu, L., Xia, Y., & Fernando, N. (2015). Drought onset mechanisms revealed by satellite solar-induced chlorophyll fluorescence: Insights from two contrasting extreme events. Journal of Geophysical Research G: Biogeosciences, 120(11), 2427–2440. https://doi.org/10.1002/2015JG003150

Turner, A. J., Köhler, P., Magney, T. S., Frankenberg, C., Fung, I., and Cohen, R. C. (2020). A double peak in the seasonality of California's photosynthesis as observed from space, Biogeosciences, 17, 405–422, https://doi.org/10.5194/bg-17-405-2020.

Veefkind, J. P., Aben, I., McMullan, K., Förster, H., De Vries, J., Otter, G., ... & Levelt, P. F. (2012). TROPOMI on the ESA Sentinel-5 Precursor: A GMES mission for global observations of the atmospheric composition for climate, air quality and ozone layer applications. Remote sensing of environment, 120, 70-83. https://doi.org/10.1016/j.rse.2011.09.027

Zhou, S., Zhang, Y., Williams, A. P., & Gentine, P. (2019). Projected increases in intensity, frequency, and terrestrial carbon costs of compound drought and aridity events. Science advances, 5(1), eaau5740. https://doi.org/10.1126/sciadv.aau5740

---

## Author Response (AR1)

**Reviewer #1**

The study analyzed the connections between different types of hydrometeorological hazards and vegetation productivity extremes. The topic is important and worth exploring considering the scenario of more intense and frequent extreme weather events. Current work relies on limited datasets and statistical approaches, while lacking more comprehensive and in-depth analysis.

We thank the reviewer for highlighting the relevance of the topic of our study. All (updated) figures and literature referred to in the answers to the reviewers questions can be found in "Supplement_on_RC.pdf". A short note on the numbering in this document: Apart from Fig. 1*, the numbering for figures in this document will be equal to the numbering in the initially submitted manuscript and the supplementary material of the manuscript. When referring to line numbers, we refer to the line numbers in the manuscript file **with** tracked changes.

My major concerns are as follows:

- Considering the uncertainties related to both SIF and re-analysis data sets, I would suggest the authors include additional data in the analysis to enhance the robustness of the work. For example, the NOAA vegetation health index (VHI) data have been widely used in monitoring global vegetation health and predicting crop yield (Kogan et al.,2004), which could be complementary to SIF in quantifying vegetation status. Similarly, the surface wetness anomalies derived from long-term satellite observations (e.g. Du et al., 2019) could serve as additional metrics to quantify extreme events.

  We thank the reviewer for the suggestions. The VHI is an additive combination of a normalized NDVI index (VCI; Kogan et al., 1995) and a normalized Land Surface Temperature (LST) Condition Index (TCI), where one parameter alpha tweaks the relative contribution of both separate indices. This alpha is usually set to 0.5, indicating equal relative contributions from VCI and TCI to VHI. VCI can change due to either energy or water stress, but as Fig. 5 c-d) shows for the Enhanced Vegetation Index (EVI, and also NDVI in the last version of the manuscript), tends to already have a bias towards water stress, as described in lines 233-238. On top of that, the VHI relies on a negative relation between NDVI and LST, which is typically not the case in energy-controlled regions in northern latitudes (Fig. 4), VHI should be used cautiously there (Karniell et al., 2006). This slight bias towards water control and questionable results in energy-controlled regions (aridity < 1) are reflected when we analyse the variables most important for VHI extremes (Fig. 1* in Supplement_on_RC.pdf) and is also the reason we chose not to include VHI in the analysis, since we intend to focus on both heat and water stress.

  Lines 233-238: "However, the overall extent of water-controlled areas is clearly larger in the case of EVI compared with the SIF results. This could (i) partly be related to the fact that EVI, being less dynamic than SIF because it is more related to vegetation greenness and structure, tends to vary at time scales more in line with that of soil moisture (Turner et al., 2020), which can support stronger correlations. Or (ii) it could be due to confounding effects of the changing soil/vegetation color between dry and wet states on the EVI signal."

However, several steps were taken to enhance robustness across different proxies for vegetation productivity: 1) we have chose to replace NDVI with EVI, as NDVI tends to saturate towards vegetation productivity maxima and 2) we included evapotranspiration (ET) from GLEAM as an extra proxy for vegetation productivity, to highlight similarities between the carbon and water cycle. Reviewer #2 asked to include ET in the analysis. As Fig. S6 shows, global distributions of hydrometeorological controls on ET extremes appear to be similar to that of SIF extremes, albeit generally slightly stronger water-control. We elaborate on these findings in lines 286-293:

"Fig. S6 illustrates similar controlling hydro-meteorological variables for SIF and evapotranspiration (ET) extremes. This suggests that carbon and water cycles are sensitive to similar hazards, which in turn enhances their impact on the land climate system via both carbon and water pathways. This further demonstrates the usefulness of SIF observations for reflecting plant transpiration (Jonard et al., 2020). Further, Fig. S6 shows that GLEAM ET extremes relate much more strongly to surface soil moisture than SIF extremes. This could be due to the part of ET that partitions into an unproductive part, bare soil evaporation, which evaporates water from the surface layer directly and a productive part, transpiration, which is connected to carbon uptake and therefore SIF. Surface soil moisture affects the unproductive part, while overall enhancing the role of surface soil moisture for ET."

Regarding the use of long-term satellite observations of soil moisture: In this analysis, we have chosen to focus on the respective role of the depth from which plants extract soil moisture. Therefore, this unfortunately excludes satellite observations of soil moisture of any kind, since they usually represent only the surface layer. Instead of satellite observed soil moisture, we use soil moisture from different depths from reanalysis data, in which station and satellite observations are being assimilated. Therefore, satellite observations of soil moisture are still indirectly represented in this analysis. To validate the robustness of the results, we re-compute the results using another soil moisture data set: SoMo.ml, on which we elaborate in lines 88-91:

"In addition to this, and to validate the robustness of our results, we use an alternative soil moisture product, SoMo.ml, which provides data for three layers (1: 0-10 cm, 2: 10-30cm, 3: 30-50cm), and which is derived through a machine learning approach that is trained with in-situ soil moisture measurements from across the globe (O and Orth, 2021)."

- It would be valuable to add analysis based on plant physiology (e.g. Porporato et al., 2001; Kunert et al., 2017) for better clarification of the inner connections between vegetation growth and hydrometeorological hazards as compared to the correlation-based analysis.
  We thank the reviewer for this comment, which alludes to better understanding biome-specific responses of plants to hydrometeorological hazards. As mentioned in Porporato et al., (2001), physiology and rooting depth modulate responses of plants to hydrometeorological hazards. We clarified this point in lines 298-306:

"To additionally explore the influence of different vegetation types and their respective plant physiological differences on the main controls of vegetation productivity, we bin the grid cell results by the respective fraction of tree cover of the entire vegetation cover, and by aridity in Fig. S7. We find that the radiation control of SIF extremes in humid regions is mostly associated with forests, and that the water control in semi-arid regions largely occurs for shorter vegetation, with presumably more shallow root systems, while productivity extremes in more forested semi-arid regions tend to be energy-controlled. In very strong droughts, tall trees with deep rooting systems are particularly prone to suffer hydraulic failure (Brum et al., 2019). However, in our analysis we consider 5 events in a 15-year time period, such that we likely don't exclusively capture very strong droughts that might result in tree mortality. Generally, hardly any changes in the most important variables can be seen with variations in tree cover, suggesting that on a global scale plant physiological differences only have a limited effect on determining the most important control for SIF extremes."

- It seems to me the lagged vegetation responses to hydrometeorological hazards and the accumulated impacts from pro-longed drought/heat wave on vegetation need to be carefully addressed. Such component is currently missing in the manuscript.
  We appreciate the point made by the reviewer. A similar point has been risen by reviewer #2. On top of response of SIF to concurrent anomalies in hydrometeorological variables, there might be lagged effects in the SIF response. To this end, Fig. S4 shows the most important hydrometeorological variables for SIF extremes in the respective following month. Patterns are comparable to Fig. 5a-b), which suggests that in the month preceding SIF extremes, energy/water deficits/surpluses are already developing. This is evidenced especially in arid regions, where precipitation and shallow soil moisture of the preceding month replace root-zone soil moisture as the most important variable. In the hottest, humid regions, preceding shallow soil moisture replaces radiation as the most important variable for the concurrent months, suggesting that water is typically abundant (SIF maximum) or lacking (SIF minima) a month ahead of the extreme. This indicates that in these regions, both energy and water should be present or lacking to obtain a SIF extreme. We have clarified this in lines 274-281:

  "Fig. S4 indicates that hydrometeorological anomalies do not solely elicit immediate, but also lagged vegetation responses. A clear difference between water- and energy-controlled conditions is already visible when correlating hydrometeorological anomalies of the preceding month with the respective SIF extreme. Energy and water surpluses and deficits establish over time, which is most clearly evidenced in arid regions, where precipitation and shallow soil moisture of the preceding month is found to be the most important variable. With time, deeper soil moisture becomes more important (Fig. 5a-b), as in case of SIF maxima, precipitation needs time to infiltrate the soil and in case of SIF minima, the soil dries most rapidly from the top down."

References:

Kogan, F., Stark, R., Gitelson, A., Jargalsaikhan, L., Dugrajav, C. and Tsooj, S., 2004. Derivation of pasture biomass in Mongolia from AVHRR-based vegetation health indices. International Journal of Remote Sensing, 25(14), pp.2889-2896.

Du, J., Kimball, J.S., Velicogna, I., Zhao, M., Jones, L.A., Watts, J.D. and Kim, Y., 2019. Multicomponent satellite assessment of drought severity in the contiguous United States from 2002 to 2017 using AMSRâ E and AMSR2. Water Resources Research, 55(7), pp.5394-5412.

Porporato, A., Laio, F., Ridolfi, L. and Rodriguez-Iturbe, I., 2001. Plants in water-controlled ecosystems: active role in hydrologic processes and response to water stress: III. Vegetation water stress. Advances in water resources, 24(7), pp.725-744.

Kunert, N., Aparecido, L.M.T., Wolff, S., Higuchi, N., dos Santos, J., de Araujo, A.C. and Trumbore, S., 2017. A revised hydrological model for the Central Amazon: The importance of emergent canopy trees in the forest water budget. Agricultural and Forest Meteorology, 239, pp.47-57.

**Reviewer #2**

This Paper "**SPATIALLY VARYING RELEVANCE OF HYDROMETEOROLOGICAL HAZARDS FOR VEGETATION PRODUCTIVITY EXTREMES**" presented very interesting research regarding the relationship between satellite-based Sun-induced chlorophyll fluorescence (SIF) as a proxy for vegetation productivity and Hydrometeorological extremities such as drought and cold spells.

We thank the reviewer for the encouraging evaluation of our paper. All (updated) figures and literature referred to in the answers to the reviewers questions can be found in "Supplement_on_RC.pdf". A short note on the numbering in this document: Apart from Fig. 1*, the numbering in this document will be equal to the numbering in the main manuscript and the supplementary material of the manuscript.

- Although this paper presents very interesting results, the overall study lacks detailed literature review. The authors claim that according to their knowledge this study is the first of its kind to analyze this relationship at a global scale. LINE 60. Such statements could have been avoided, there might be several studies discussing such topics, for example (Jonard et al., 2020) (Sun et al., 2015).

  We thank the reviewer for the literature suggestions. The claim we make in this study is that to our knowledge it is the first time that vegetation productivity extremes as inferred from satellite-observed SIF are being analyzed alongside the occurrence of single or compound hazards. To more explicitly state the novelty of this study, we have adjusted lines 59-63, and incorporated literature suggested by the reviewer:

  "In this study, we re-visit the relationship between vegetation productivity and hydrometeorological hazards by analyzing the implications of both single and compound hazards on vegetation productivity extremes, as has been highlighted before (Sun et al., 2015, Zhou et al., 2019). However, to our knowledge for the first time, we do so comprehensively by approximating variable importance during vegetation productivity extremes inferred from SIF data on a global scale."

- The authors also did not include any statistical analysis to validate their findings and explain how accurate their acquired results are? Statistical analysis of the acquired results and hydrometeorological data could have been done to evaluate/validate their claim.

  We perform bootstrapping, and we consider p-values of the inferred correlations to validate the robustness of our obtained results. In particular, we perform the following analyses:

  1) To verify whether hydrometeorological anomalies during SIF extremes are actually hazardous, we (i) randomly sample 5 temperature and soil moisture anomalies from the data with sufficiently active vegetation and average these anomalies. (ii) We repeat this step 100 times to acquire a bootstrapped sample of temperature and soil moisture anomalies during normal conditions, from which we determine the 10th and 90th percentile. (iii) We only classify a hydrometeorological hazard when temperature and soil moisture anomalies are below the 10th (cold spell or drought) or above the 90th (heat wave or wet spell) percentile.
  2) To assure statistically meaningful relationships, we focus only on statistically significant correlations (p-values < 0.05).

  The measures taken to ensure statistical robustness of the methodology are clarified in lines 125-144 and 149-151.

  "Then, a series of steps is taken to test if the coinciding hydrometeorological anomalies during SIF extremes are actually hazardous: (i) We randomly sample five months with sufficiently active vegetation and average the soil moisture and temperature anomalies, respectively, across them. (ii) We repeat this 100 times to obtain a distribution from which we determine the 10th and 90th percentile. (iii) A hydrometeorological hazard is detected if the actual, averaged temperature and/or soil moisture anomalies associated with the SIF extremes are below 10th (cold spell or drought) or above the 90th percentile (heat wave or wet spell) of the distribution of randomly sampled averaged anomalies. Note that with this approach we can detect both single and compound hydrometeorological hazards.

  Complementing this analysis, in the second approach we analyze the temporal co-variation between SIF extremes and hydrometeorological anomalies. For this purpose, we correlate the five SIF extreme anomalies with anomalies of all considered hydrometeorological variables in each grid cell. We include respective SIF and hydrometeorological data from the surrounding grid cells to yield a larger data sample consisting of 5 x (8+1) = 45 data pairs. We disregard negative and insignificant (p-value > 0.05) correlations, as we assume these are not indicating actual physical controls but rather represent the influence of noise or confounding effects such as low precipitation during times of high radiation."

  "Finally, the hydrometeorological variable that yields the highest correlation coefficient with the extreme SIF anomalies is regarded as the main SIF-controlling variable during vegetation productivity maxima or minima."

Next to this, we emphasize the role of the applied statistical measures in lines 144-148:

"This also serves to deal with uncertainty in the SIF data set. When systematic patterns emerge from either of the approaches with adequate significance, they are unlikely confounded by underlying SIF patterns: as we focus solely on either SIF maxima or minima, statistically significant relations only emerge when concurrent hydrometeorological anomalies of an appropriate magnitude exist."
- Several papers are discussing spatial scale dependencies of SIF, its accuracy and biases. Authors could have extended the discussion to illustrate how their GOME SIF data justify its relationship with meteorological hazards.

  Assessing the scale dependency between GOME-2 SIF data and vegetation processes that are occurring locally is vital for the usefulness of this study. To clarify this, we elaborated on scale dependencies in lines 374-382:

  "Our results are obtained at, and valid for, relatively large spatial (half degree) and temporal (monthly) scales. Previous studies have shown differences in the vegetation-climate coupling across scales (Linscheid et al., 2020), suggesting it would be worthwhile to repeat our analysis for different spatiotemporal scales in the future, possibly with new satellite data products. In this context it should be noted, however, that while the relationship between SIF and gross primary productivity (GPP) as actual vegetation productivity is strong for large spatio-temporal scales (Frankenberg et al., 2011; Guanter et al., 2012; Joiner et al., 2013), it can deteriorate towards smaller scales (He et al., 2020; Maguire et al., 2020; Marrs et al., 2020; Wohlfahrt et al., 2018). And the spatiotemporal range within which there is an acceptable SIF-GPP relationship is not entirely clear yet."
- It is also recommended to include GOSAT data to make this research more accurate (Guanter et al., 2012). This paper (Frankenberg et al., 2011) which authors have also cited discussed GOSAT findings so it is suggested to include GOSAT analysis as well. Also, there are a couple of reconstructed long term time series of SIF, which are available.

  We thank the reviewer for the suggestions. In the updated version of the manuscript, our findings are supported by a versatile array of indices representing different perspectives on vegetation (SIF, EVI and ET). Moreover, because SIF from GOSAT has been found to be much more noisy than GOME-2 SIF (Joiner et al., 2013), we chose not to include SIF from GOSAT in this analysis.

  Concerning the reconstructed SIF time series: While we acknowledge that longer time series can improve the signal to noise ratio in our analysis, we expect that reconstructed time series will i) include additional uncertainty compared to the observation based products and ii) introduce confounding patterns, which can again deteriorate the signal to noise ratio and complicate the interpretation of the results. Therefore, we prefer to avoid the inclusion of reconstructed products in our analysis.

- Authors could have included EVI for their analysis, as the NDVI might get saturated over the dense canopies., NDVI saturates under high biomass and is unable to replicate the actual biomass content of the vegetation. EVI is considered better among other vegetation indices for the identification of vegetating content

(Bandopadhyay et al., 2021). Another very good paper discussing SIF and EVI relationship in Africa is (Getachew Mengistu et al., 2021).

We fully agree with the reviewer and have replaced all NDVI related results throughout the manuscript and the supplementary material with respective EVI analyses (see for example revised Fig. 5c-d), S5, and S7, and adapted lines 226-237). As can be seen from Fig. 5c-d), this does not change our results significantly, suggesting that, on a global scale, NDVI saturation plays a negligible role on the timing of the vegetation productivity extreme.

Lines 226-237:
"Furthermore, we repeat our co-variability analysis for EVI instead of SIF in Fig. S5, which allows us to contrast to some extent the behavior of vegetation physiology (SIF) and vegetation structure (EVI). Similar to the spatial patterns of energy- and water-controlled vegetation in Fig. 4, EVI shows predominant energy control at high latitudes, while the mid latitudes are largely water-controlled. Further, as in Fig. 4 for SIF, EVI minima are more associated with water variables than EVI maxima.
However, the overall extent of water-controlled areas is clearly larger in the case of EVI compared with the SIF results. This could (i) partly be related to the fact that EVI, being less dynamic than SIF because it is more related to vegetation greenness and structure, tends to vary at time scales more in line with that of soil moisture (Turner et al., 2020), which can support stronger correlations. Or (ii) it could be due to confounding effects of the changing soil/vegetation color between dry and wet states on the EVI signal."

- Although this article deals with the hydrometeorological hazards and SIF there is no data displayed or discussion regarding the flux towers or evapotranspiration. Authors should include some literature related to its relationship with both SIF and meteorological phenomenon (Zuromski et al., 2018).

We agree with the reviewer that highlighting the similarities between extremes in the water and carbon cycle increases the scientific value of the findings. To stress similarities between SIF and ET extremes, we re-computed our analysis using GLEAM ET. Global distributions of hydrometeorological controls on ET extremes appear to be similar to that of SIF extremes, albeit generally slightly stronger water-control. Moreover, going from ET maxima to minima, a similar transition from energy to water control can be seen in semi-arid regions (1 < aridity < 2). In general, GLEAM ET correlates much more with surface soil moisture. This could be due to the part of ET that partitions into bare soil evaporation, which evaporates water from the surface layer directly, therefore enhancing the role of surface soil moisture for ET. We elaborate on these findings in lines 286-293:

"Fig. S6 illustrates similar controlling hydro-meteorological variables for SIF and evapotranspiration (ET) extremes. This suggests that carbon and water cycles are sensitive to similar hazards, which in turn enhances their impact on the land climate system via both carbon and water pathways. This further demonstrates the usefulness of SIF observations for reflecting plant transpiration (Jonard et al., 2020). Further, Fig. S6 shows that GLEAM ET extremes relate much more strongly to surface soil moisture than SIF extremes. This could be due to the part of ET that partitions into an unproductive part, bare soil evaporation, which evaporates water

from the surface layer directly and a productive part, transpiration, which is connected to carbon uptake and therefore SIF. Surface soil moisture affects the unproductive part, while overall enhancing the role of surface soil moisture for ET."

- Although this paper deals with the time 2007 to 2015. The OCO-2, launched on July 2, 2014, has SIF Product with fine detail and a better retrieval algorithm. Authors should incorporate OCO-2 data to make their results more useful (Frankenberg et al., 2014) and advancement of knowledge and methods.
  We agree with the reviewer that doing this analysis with higher-quality OCO-2 data would be interesting. However, because our methodology is based on anomalies, we require time series that are sufficiently long to calculate robust monthly climatologies and corresponding deviations. This unfortunately puts including OCO-2, or any other higher resolution SIF measurements from TROPOMI (Koehler et al., 2018, Veefkind et al., 2012), out of scope for this analysis.

- The authors also missed an important aspect of vegetation response to climate extreme which might be better represented with a lag time.
  We appreciate the point made by the reviewer. A similar point has been risen by reviewer #1. On top of response of SIF to concurrent anomalies in hydrometeorological variables, there might be lagged effects in the SIF response. To this end, Fig. S4 shows the most important hydrometeorological variables for SIF extremes in the respective following month. Patterns are comparable to Fig. 5a-b), which suggests that in the month preceding SIF extremes, energy/water deficits/surpluses are already developing. This is evidenced especially in arid regions, where precipitation and shallow soil moisture of the preceding month replace root-zone soil moisture as the most important variable. In the hottest, humid regions, preceding shallow soil moisture replaces radiation as the most important variable for the concurrent months, suggesting that water is typically abundant (SIF maximum) or lacking (SIF minima) a month ahead of the extreme. This indicates that in these regions, both energy and water should be present or lacking to obtain a SIF extreme. We have clarified this in lines 274-281:

  "Fig. S4 indicates that hydrometeorological anomalies do not solely elicit immediate, but also lagged vegetation responses. A clear difference between water- and energy-controlled conditions is already visible when correlating hydrometeorological anomalies of the preceding month with the respective SIF extreme. Energy and water surpluses and deficits establish over time, which is most clearly evidenced in arid regions, where precipitation and shallow soil moisture of the preceding month is found to be the most important variable. With time, deeper soil moisture becomes more important (Fig. 5a-b), as in case of SIF maxima, precipitation needs time to infiltrate the soil and in case of SIF minima, the soil dries most rapidly from the top down.

References

Bandopadhyay, S., Rastogi, A., Cogliati, S., Rascher, U., GÄ...bka, M., & Juszczak, R. (2021). Can vegetation indices serve as proxies for potential sun-induced fluorescence (SIF)? A fuzzy simulation approach on airborne imaging spectroscopy data. Remote Sensing, 13(13), 1–22. https://doi.org/10.3390/rs13132545

Frankenberg, C., Fisher, J. B., Worden, J., Badgley, G., Saatchi, S. S., Lee, J.-E., Toon, G. C., Butz, A., Jung, M., Kuze, A., & Yokota, T. (2011). New global observations of the terrestrial carbon cycle from GOSAT: Patterns of plant fluorescence with gross primary productivity. Geophysical Research Letters, 38(17), n/a-n/a. https://doi.org/10.1029/2011GL048738

Frankenberg, C., O'Dell, C., Berry, J., Guanter, L., Joiner, J., Köhler, P., Pollock, R., & Taylor, T. E. (2014). Prospects for chlorophyll fluorescence remote sensing from the Orbiting Carbon Observatory-2. Remote Sensing of Environment, 147, 1–12. https://doi.org/10.1016/j.rse.2014.02.007

Getachew Mengistu, A., Mengistu Tsidu, G., Koren, G., Kooreman, M. L., Folkert Boersma, K., Tagesson, T., Ardö, J., Nouvellon, Y., & Peters, W. (2021). Sun-induced fluorescence and near-infrared reflectance of vegetation track the seasonal dynamics of gross primary production over Africa. Biogeosciences, 18(9), 2843–2857. https://doi.org/10.5194/bg-18-2843-2021

Guanter, L., Frankenberg, C., Dudhia, A., Lewis, P. E., Gómez-Dans, J., Kuze, A., Suto, H., & Grainger, R. G. (2012). Retrieval and global assessment of terrestrial chlorophyll fluorescence from GOSAT space measurements. Remote Sensing of Environment, 121, 236–251. https://doi.org/10.1016/j.rse.2012.02.006

Jonard, F., De Cannière, S., Brüggemann, N., Gentine, P., Short Gianotti, D. J., Lobet, G., Miralles, D. G., Montzka, C., Pagán, B. R., Rascher, U., & Vereecken, H. (2020). Value of sun-induced chlorophyll fluorescence for quantifying hydrological states and fluxes: Current status and challenges. Agricultural and Forest Meteorology, 291(June), 108088. https://doi.org/10.1016/j.agrformet.2020.108088

Sun, Y., Fu, R., Dickinson, R., Joiner, J., Frankenberg, C., Gu, L., Xia, Y., & Fernando, N. (2015). Drought onset mechanisms revealed by satellite solar-induced chlorophyll fluorescence: Insights from two contrasting extreme events. Journal of Geophysical Research G: Biogeosciences, 120(11), 2427–2440. https://doi.org/10.1002/2015JG003150

Zuromski, L. M., Bowling, D. R., Köhler, P., Frankenberg, C., Goulden, M. L., Blanken, P. D., & Lin, J. C. (2018). Solar-Induced Fluorescence Detects Interannual Variation in Gross Primary Production of Coniferous Forests in the Western United States. In Geophysical Research Letters (Vol. 45, Issue 14, pp. 7184–7193). https://doi.org/10.1029/2018GL077906